# Neutrophils at the Crossroads of Oral Microbiome Dysbiosis and Periodontal Disease

**DOI:** 10.3390/microorganisms13112573

**Published:** 2025-11-11

**Authors:** João Viana, Tiago Ferro, Ricardo Pitschieller, Vanessa Machado, Naichuan Su, José João Mendes, João Botelho

**Affiliations:** 1Egas Moniz Center for Interdisciplinary Research, Egas Moniz School of Health and Science, Quinta da Granja, Campus Universitário, Monte da Caparica, 2829-511 Almada, Portugal; tferro@egasmoniz.edu.pt (T.F.); r.pit@oraldesign.com.pt (R.P.); vmachado@egasmoniz.edu.pt (V.M.); jmendes@egasmoniz.edu.pt (J.J.M.); jbotelho@egasmoniz.edu.pt (J.B.); 2Department of Oral Public Health, Academic Center for Dentistry Amsterdam (ACTA), University of Amsterdam and Vrije Universiteit Amsterdam, 1081 LA Amsterdam, The Netherlands; n.su@acta.nl

**Keywords:** neutrophil extracellular traps (NETs), phagocytosis, chemotaxis, periodontal disease, oral microbiome, biomarkers, therapeutic strategies

## Abstract

Neutrophils are the most abundant circulating leukocytes and essential components of innate immunity. Through mechanisms such as phagocytosis, reactive oxygen species (ROS) production, degranulation, and neutrophil extracellular trap (NET) formation, they play a crucial role in host defense. However, dysregulated neutrophil responses are linked to chronic inflammatory conditions, including periodontitis. This review summarizes current evidence on neutrophil biology in periodontal health and disease, focusing on functional mechanisms, recruitment pathways, the influence of dysbiosis, and their potential as biomarkers and therapeutic targets. Neutrophils display a dual role in periodontal tissues: while protecting against microbial invasion, their excessive or impaired activity contributes to tissue destruction. Altered chemotaxis, defective phagocytosis, and uncontrolled NET release perpetuate inflammation and alveolar bone loss. Neutrophil-derived enzymes, including myeloperoxidase, elastase, and matrix metalloproteinases, emerge as promising biomarkers for early diagnosis. In parallel, therapeutic strategies targeting oxidative stress, NET regulation, or neutrophil hyperactivity are being explored to preserve periodontal tissues. Neutrophils are central players in periodontal pathophysiology. Understanding their regulation and interaction with the oral microbiome may enable the development of novel diagnostic and therapeutic approaches, ultimately improving periodontal disease management.

## 1. Introduction

Neutrophils are polymorphonuclear leukocytes (PMNs) that play a fundamental role in the body’s innate immune defense [1]. They represent the most abundant population of circulating leukocytes [2]. These cells exhibit a short lifespan within the bloodstream, typically ranging from a few hours to 5 days, on average, and serve as the first line of defense, particularly against extracellular pathogenic microorganisms [3].

Produced in the bone marrow from myeloid progenitor cells, they are continuously released into the circulation. Less mature neutrophils can also be released in response to acute inflammatory or infectious stimuli, being activated once arriving at the site of injury [4]. There, neutrophils eliminate pathogens through various mechanisms, including phagocytosis, the release of antimicrobial enzymes, the production of reactive oxygen species (ROS) and the formation of neutrophil extracellular traps (NETs) as shown in Figure 1 [5]. These processes contain and destroy invading agents [6]; however, when dysregulated or impaired, they can lead to tissue damage and exacerbation of inflammatory responses [7]. These impaired physiological pathways are observed in chronic inflammatory diseases, including periodontitis [8].

Human neutrophils can be reliably identified and characterized based on a distinct pattern of surface marker expression. Mature circulating neutrophils typically express high levels of CD15 and CD66b, which are associated with granulocytic lineage and activation status [9,10]. CD16 (FcγRIIIb) and CD11b (integrin αM) are also highly expressed and play crucial roles in antibody-dependent functions and adhesion, respectively. Additionally, CD10 can be used to distinguish mature neutrophils from immature forms, such as band cells or myelocytes. In contrast, human neutrophils lack expression of hematopoietic stem and progenitor cell markers, such as CD34, and do not express major histocompatibility complex (MHC) class II molecules [11]. This specific combination of surface marker expression enables the discrimination of neutrophils from other immune cell populations and facilitates their functional and developmental analysis in both physiological and pathological contexts.

Neutrophils are central to immune defence eliminating microorganisms mainly by phagocytosis [12]. This process begins with the recognition of pathogens by receptors such as Toll-like receptors (TLRs), Fc and complement. The internalization of extracellular pathogens or particles leads to the formation of the phagolysosome, where proteolytic enzymes, ROS and antimicrobial peptides destroy the microorganisms [13]. An additional mechanism of neutrophil activity includes the formation of NETs, in a process termed NETosis. These NETs are composed of DNA, histones and antimicrobial proteins, an effective strategy against infections, but which, when deregulated, can contribute to chronic inflammation and tissue destruction [14].

Recruitment of neutrophils to infection sites occurs by chemotaxis, following gradients of cytokines and chemokines such as IL-8 and C5a [5]. This process begins with rolling and adhesion to the endothelium, mediated by selectins. Endothelial activation leads to increased vascular permeability, facilitating neutrophil migration and diapedesis, mediated by integrins such as LFA-1 and Mac-1 [12]. Once in the inflamed tissue, neutrophils eliminate microorganisms and modulate the immune response, but their excessive or prolonged activity can contribute to the persistence of chronic inflammation, as seen in periodontitis [15]. Normally, apoptosis and removal of these cells guarantee resolution of the inflammation, but dysfunctions in this mechanism can lead to prolonged tissue damage [16].

Neutrophils are essential components of the innate immune system, serving as the first line of defense against pathogens [9,17]. These polymorphonuclear leukocytes employ various mechanisms to combat invading microorganisms, including phagocytosis, production of ROS, and release of proteolytic enzymes [17,18]. A recently discovered defense strategy is the formation of NETs), which are extracellular structures composed of DNA and antimicrobial proteins that capture and neutralize pathogens [2,19]. While NETs play a crucial role in host defense, excessive formation or impaired clearance can lead to tissue damage and contribute to autoimmune diseases [20]. Neutrophils also participate in the resolution of inflammation and interact with the adaptive immune system, highlighting their multifaceted role in immune responses [9,19].

Emerging clinical and experimental evidence seem to fairly indicate a neutrophil hyperresponsiveness in periodontitis that may contribute to systemic low-grade inflammation [1]. In individuals with genetic or epigenetic susceptibility, such as polymorphisms in genes regulating neutrophil chemotaxis and activation (e.g., FPR1, CXCR1/2, TLR4), the persistent stimulation by dysbiotic biofilms maintains neutrophils in a primed state [2]. This ultimately results in exaggerated production of reactive oxygen species and enhanced formation of NETs. Epigenetic alterations, including DNA methylation changes in promoters of inflammatory genes, as well as behavioural factors such as smoking or high-sugar diets, further augment this priming and delay resolution of inflammation [1,2].

Once activated, neutrophils release large quantities of elastase, myeloperoxidase, and other proteolytic and oxidative mediators [3]. These molecules can damage endothelial cells, reduce nitric oxide bioavailability, and increase vascular permeability, thereby promoting a systemic pro-inflammatory milieu [4]. While a causal relationship between periodontitis-induced neutrophil dysregulation and systemic disease has not yet been definitively established, neutrophil hyperactivity and excessive NETosis are consistently reported in diabetes mellitus, rheumatoid arthritis, and inflammatory bowel disease. The recurrence of this neutrophil-driven inflammatory signature across conditions suggests that altered neutrophil function may represent a shared pathophysiological mechanism linking periodontal and systemic inflammation [1].

Neutrophils play a crucial role in maintaining oral health and limiting the development of periodontitis, a chronic inflammatory disease affecting tooth-supporting tissues [16]. These immune cells survey the local environment and can be activated by cues provided by dysbiotic bacteria. Key mechanisms of neutrophil activity include the respiratory burst, formation of NETs, degranulation, and phagocytosis [21]. However, neutrophils exhibit a dual role in periodontitis, contributing to both protective and destructive processes [22]. There is a well-maintained equilibrium in oral tissues, as impaired neutrophil activity or reduced numbers of these cells can lead to bacterial overgrowth and tissue damage, while their excessive activity can cause bystander injury through excessive inflammatory responses [16]. The interaction between neutrophils and the oral microbiome is complex, with a shift from a symbiotic to a dysbiotic bacterial community often underlying the development of periodontitis [23]. Understanding these intricate relationships is crucial for developing novel therapeutic approaches while keeping a balance between homeostatic immunity and inflammatory pathology in periodontal health [16,22].

Neutrophils play a crucial role in periodontal health and disease, acting as both defenders against pathogens and potential contributors to tissue destruction [5]. While their absence or dysfunction can lead to severe periodontitis, excessive neutrophil activity can also result in inflammatory tissue breakdown [24]. Neutrophils form a protective barrier against bacterial invasion but may persist in periodontal tissues due to bacterial evasion mechanisms and delayed neutrophil clearance [25]. Their antimicrobial functions, including the recently discovered NETs, are essential for innate immunity but can also contribute to tissue damage [26]. The complex role of neutrophils in periodontitis involves a delicate balance between host defense and potential tissue destruction, making them both critical for protection and possible perpetrators of disease progression [5,25].

## 2. What Is the Possible Relationship Between Neutrophils and Periodontal Disease?

The involvement of neutrophils is central to periodontal health as well as disease development, maintaining homeostasis through their antimicrobial functions [5]. These cells form a barrier against bacteria in the gingiva, but their absence or dysfunction can lead to severe periodontitis [27]. Paradoxically, both neutrophil deficiency and hyperactivity are associated with periodontal disease progression [26]. Neutrophils possess various antimicrobial mechanisms, including phagocytosis and the formation of NETs [28]. However, their antimicrobial actions can also contribute to tissue destruction in periodontitis [29]. The balance between neutrophil function and microbial challenge is crucial for maintaining periodontal health [24]. Recent research has revealed that some oral bacteria can subvert neutrophil responses, promoting dysbiosis and inflammatory tissue breakdown [5]. These findings are informing the development of novel therapeutic approaches for periodontal disease treatment [5,27].

NETs play a crucial role in periodontal health and disease. In healthy conditions, NETs help maintain homeostasis by trapping and eliminating pathogens [30]. However, in periodontitis, dysbiotic microbial communities trigger excessive NET formation, contributing to tissue destruction [8]. NETs consist of extruded DNA, histones, and antimicrobial peptides, which can effectively combat various microorganisms but may also cause host tissue damage [8]. The pathogenesis of periodontitis involves impaired NET formation or elimination, leading to exacerbated inflammation and gingival tissue destruction [31]. Periodontal pathogens have developed mechanisms to resist NETs, including DNA breakdown and degradation of antibacterial proteins [8]. Potential therapeutic approaches for periodontitis include regulating NET concentrations, local anti-inflammatory therapy, and targeted antibacterial treatments [32]. NET biomarkers may prove useful in diagnosing periodontitis, but further research is needed to elucidate specific NET mechanisms and bacterial interactions [30].

At the same time, there is evidence that in some individuals susceptible to periodontitis neutrophils may show functional defects, including deficiencies in chemotaxis and phagocytosis.

Neutrophil dysfunction can lead to several primary immunodeficiencies, the most common of which include primary neutropenia, chronic granulomatous disease (CGD), and leukocyte adhesion defects (LAD) [28,33]. Primary neutropenia is characterised by a persistent reduction in circulating neutrophil count, which impairs innate immune responses and increases susceptibility to recurrent bacterial and fungal infections [34,35]. Chronic granulomatous disease (CGD) results from mutations affecting the NADPH oxidase complex, which is essential for producing ROS in phagocytes, leading to an inability to kill ingested pathogens [25,36]. Leukocyte adhesion defects (LAD) involve mutations in cellular adhesion molecules, such as integrins, preventing neutrophils from migrating effectively to sites of infection [23,37]. These disorders highlight the critical role of neutrophils in host defence and the severe consequences of their genetic impairment [22,35].

These alterations result in an inability to effectively eliminate periodontopathogenic microorganisms, allowing the infection to persist and contributing to the progression of the disease [26,38]. Genetic diseases such as type I leukocyte adhesion deficiency (*LAD-I*) illustrate this phenomenon, as individuals affected by this condition have a compromised immune response and develop early aggressive periodontitis due to neutrophil dysfunction [39].

The transition from gingivitis to periodontitis is another process in which neutrophils play a decisive role [40]. During gingivitis, neutrophil infiltration of the gums occurs as an initial response to bacterial biofilm, partially controlling the infection [15]. However, when local inflammatory processes become persistent and the innate immune response is unable to eliminate the pathogenic stimulus, the balance between protective and destructive mechanisms is disrupted, ultimately leading to the degradation of periodontal tissues and alveolar bone loss characteristic of periodontitis [41].

Given the significant impact of neutrophil involvement in the pathogenesis of periodontitis, various therapeutic approaches have been proposed to modulate the immune response, with the objective of minimizing collateral tissue damage as presented in Figure 2 [21]. The use of matrix metalloproteinase (MMP) inhibitors, such as doxycycline in low doses (subantimicrobial), has shown efficacy in reducing connective tissue destruction without compromising immune function [21]. Similarly, interventions aimed at reducing oxidative stress, such as the administration of antioxidants, can mitigate the adverse effects of excessive ROS production by neutrophils [42,43]. In addition, recent research into the regulation of NETs suggests that they could be a therapeutic target, since their uncontrolled formation contributes to chronic inflammation in periodontitis [26].

## 3. What Are the Possible Mechanisms of Neutrophil Recruitment in Periodontitis?

The recruitment of neutrophils to periodontal tissues is a fundamental process in the innate immune response, guaranteeing the elimination of pathogens and the maintenance of tissue homeostasis [5]. Neutrophil migration to periodontal tissues is mediated by a complex system of chemotactic signals, cellular interactions and changes in the inflammatory microenvironment, which direct these cells from the vascular compartment to the site of infection [24].

The initial activation of the inflammatory response in periodontal tissues occurs due to the detection of microbial components by the innate immune system [27]. The presence of bacterial biofilms, composed of periodontopathogenic microorganisms such as *Porphyromonas gingivalis* and *Aggregatibacter actinomycetemcomitans*, induces the expression of Pathogen-Associated Molecular Patterns (PAMPs), which are recognised by Toll-like receptors (TLRs) present on epithelial cells, macrophages and dendritic cells [44]. Activation of these receptors triggers an inflammatory signalling cascade that culminates in the production of pro-inflammatory cytokines, including interleukin-8 (IL-8), tumour necrosis factor-alpha (TNF-α), interleukin-1 beta (IL-1β) and prostaglandin E_2_ (PGE_2_) [45]. These mediators promote the recruitment of neutrophils by creating a chemotactic gradient directed towards the site of infection [21].

Following activation of the inflammatory response, circulating neutrophils respond rapidly to chemotactic signals, initiating the process of migration to the periodontal tissues [46]. This process takes place in several highly regulated stages. Initially, neutrophils interact with the vascular endothelium by binding glycoproteins to adhesion molecules expressed by endothelial cells, such as selectins (E-selectin and P-selectin), allowing these cells to roll along the blood vessel wall as illustrated in Figure 3 [47,48]. As recruitment progresses, integrins present on the surface of neutrophils, such as LFA-1 (CD11a/CD18) and Mac-1 (CD11b/CD18), bind strongly to molecules of the endothelial immunoglobulin family, such as ICAM-1 and ICAM-2, promoting firm adhesion of the cells to the endothelium [5]. This interaction is essential for neutrophils to cross the endothelial barrier in a process known as diapedesis, mediated by proteins such as PECAM-1 (CD31) and CD99 [46,47].

Once in the periodontal tissues, neutrophils continue their migration guided by chemotactic gradients formed by inflammatory mediators and lipid metabolites such as leukotrienes (LTB_4_) and complement C5a [15]. However, in periodontitis, this recruitment can be deregulated. In individuals susceptible to periodontal disease, a significant increase in neutrophil infiltration in gingival tissues has been observed, a phenomenon that is associated with an exacerbated immune response and increased tissue destruction [41]. This phenomenon, known as neutrophil hyperchemotaxis, contributes to the perpetuation of the chronic inflammatory state characteristic of periodontitis [21,40].

Some forms of periodontitis, particularly aggressive periodontitis, are associated with defects in the chemotactic function of neutrophils [26,49]. Studies show that, in certain patients, neutrophils have a reduced ability to respond to chemotactic gradients, jeopardizing efficient migration to the site of infection [49]. Deficiencies in chemokine receptors, such as CXCR1 and CXCR2, have been identified in some individuals with severe periodontitis, suggesting that genetic or epigenetic alterations may modulate the effectiveness of neutrophil recruitment and influence the progression of the disease [44]. This deficiency in the immune response allows periodontopathogenic microorganisms to persist in the dental biofilm, contributing to worsening inflammation and tissue destruction [41].

In addition to the response mediated by cytokines and chemokines, neutrophil recruitment in periodontitis is influenced by microbial dysbiosis [44]. The altered ecological balance of the oral microbiota, characterized by an increase in pathogenic species and a decrease in beneficial bacteria, promotes a deregulated activation of the innate immune response [40]. *Porphyromonas gingivalis*, one of the main pathogens associated with periodontitis, has immune evasion mechanisms that modulate neutrophil activity, allowing it to survive in the periodontal environment [45]. This bacterium interferes with Toll receptor signaling and inhibits the phagocytic capacity of neutrophils, prolonging inflammation and favoring tissue destruction [5].

Given the importance of neutrophil recruitment in the pathophysiology of periodontitis, various therapeutic approaches have been explored with the aim of modulating this inflammatory response [50]. The use of chemokine inhibitors has been proposed as a strategy to reduce the excessive infiltration of neutrophils into periodontal tissues, minimizing the tissue destruction associated with the disease [5]. In addition, the administration of selective anti-inflammatory drugs, such as IL-1β inhibitors or PGE_2_ modulators, may offer benefits by controlling inflammation without compromising the host’s ability to eliminate pathogens [5].

Another promising approach consists of modulating the oral microbiota through the use of probiotics and dietary interventions [23]. Promoting the growth of beneficial bacteria and reducing the pathogenic microbial load can contribute to lower neutrophil recruitment and a more balanced immune response [31]. In parallel, recent research has explored the role of NETs in periodontitis, since their excessive formation has been associated with increased inflammation and tissue destruction [40].

The recruitment of neutrophils to periodontal tissues is therefore a highly dynamic and complex process, regulated by multiple immunological and microbial factors [8]. While in normal conditions this response is controlled and efficient in eliminating microorganisms, in periodontitis, the hyperactivity or chemotactic dysfunction of neutrophils contributes to the perpetuation of chronic inflammation and the progressive destruction of periodontal tissues [16]. Continued research into the mechanisms involved in this process could open up new therapeutic perspectives, allowing for the development of more targeted approaches to controlling inflammation and preserving the integrity of the periodontium [5].

## 4. Can Neutrophils Act as Indicators of the Periodontal State?

Early identification of periodontitis is fundamental for implementing effective therapeutic strategies that prevent the progressive destruction of periodontal tissues. Traditionally, the diagnosis of periodontal disease is based on clinical parameters such as probing depth, clinical attachment loss and gingival bleeding, as well as radiographic examinations to assess bone resorption. However, these methods have limitations, as they reflect already established alterations in the tissues and do not provide detailed information on ongoing inflammatory activity. In this context, neutrophils have emerged as potential diagnostic markers for periodontitis, due to their central role in the inflammatory response and their abundant presence in the gingival sulcus and saliva.

Neutrophil infiltration of periodontal tissues is one of the immune system’s first responses to bacterial colonization, and its activity can be used to distinguish between a state of health and periodontal disease. Studies have shown that individuals with periodontitis show a significant increase in the number of neutrophils in gingival crevicular fluid (GFCF) and saliva, reflecting the exacerbated immune response to bacterial infection [51,52].

In addition to the cell count, analyzing the functional activity of neutrophils can provide valuable information on the progression of the disease. Among the main biomarkers that have been studied for the diagnosis of periodontitis are proteolytic enzymes released by neutrophils, such as myeloperoxidase (MPO) and MMPs. MPO is an enzyme released by activated neutrophils involved in the production of ROS, playing an essential role in the destruction of microorganisms. However, high levels of MPO in gingival crevicular fluid are associated with increased oxidative stress and tissue degradation, making this enzyme a possible indicator of deleterious inflammatory activity in periodontitis [53].

Similarly, neutrophil elastase, a protease responsible for the degradation of extracellular matrix proteins, has been found in significantly higher concentrations in periodontitis patients compared to healthy individuals, suggesting its potential as a diagnostic marker [54].

Matrix metalloproteinases, particularly MMP-8 and MMP-9, have also been widely investigated as markers of neutrophil-mediated tissue destruction. MMP-8, also known as neutrophil collagenase, is one of the main enzymes involved in the degradation of type I collagen, an essential component of the periodontal ligament [55]. Studies show that MMP-8 levels in gingival crevicular fluid are significantly higher in patients with active periodontitis, correlating with the severity of the disease [56]. For this reason, diagnostic tests based on the quantification of MMP-8 in saliva and GFCF have been developed, allowing non-invasive detection of periodontal inflammatory activity [57].

Another relevant parameter in assessing the neutrophil response in periodontitis is NET production [58]. The deregulated formation of NETs has been implicated in the excessive inflammation observed in periodontitis, and some studies suggest that the quantification of NETs, using the immunofluorescence technique in gingival crevicular fluid, may serve as an additional biomarker of disease progression [59].

In addition to biochemical analyses, tests that assess neutrophil chemotaxis and phagocytosis have been proposed to identify individuals with greater susceptibility to periodontitis [60]. Alterations in the migratory capacity and phagocytic efficacy of neutrophils have been observed in patients with aggressive periodontitis, suggesting that these tests could play a complementary role in stratifying the risk of the disease [61].

The use of neutrophils as diagnostic biomarkers for periodontitis represents a significant advance in the clinical approach to the disease [62]. Methods based on the quantification of neutrophil enzymes in gingival crevicular fluid and saliva offer the advantage of being minimally invasive, allowing for more precise monitoring of the inflammatory response in real time [63]. Although more studies are needed to validate these biomarkers on a large scale, current evidence suggests that assessing neutrophil function and activity can provide valuable information for early diagnosis and for personalizing therapeutic strategies in periodontitis [64].

## 5. Conclusions

Neutrophils are central to periodontal health, acting as key effectors of innate immunity while also driving tissue destruction when dysregulated. Their dual role reflects a delicate balance between protective and pathogenic functions, influenced by microbial dysbiosis and host susceptibility. Emerging evidence supports the use of neutrophil-derived biomarkers for early diagnosis and monitoring of periodontitis, while therapeutic strategies targeting neutrophil activity hold promise for restoring immune balance and preserving periodontal tissues. Moreover, growing evidence suggests that neutrophil dysfunction and chronic periodontal inflammation may contribute to systemic conditions such as diabetes mellitus, rheumatoid arthritis, and inflammatory bowel disease, highlighting the broader clinical relevance of neutrophil regulation in periodontal disease.

## Figures and Tables

**Figure 1 microorganisms-13-02573-f001:**
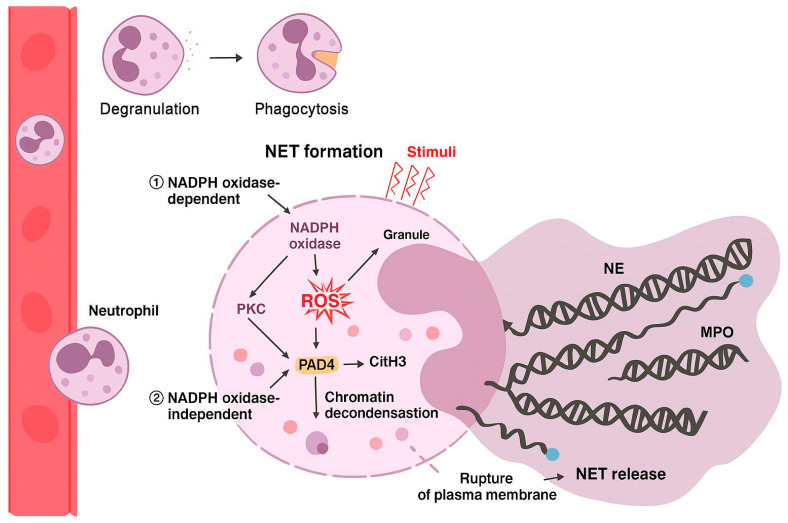
NET formation by neutrophils. Stimuli activate NADPH oxidase-dependent or -independent pathways, leading to ROS production, PAD4 activation, chromatin decondensation, and extracellular release of DNA bound to granular proteins (NE, MPO) following plasma membrane rupture.

**Figure 2 microorganisms-13-02573-f002:**
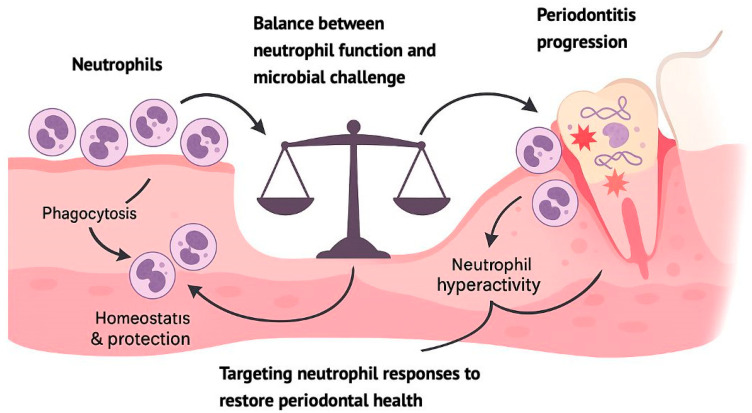
Neutrophil homeostasis preserves periodontal health, whereas hyperactivity contributes to tissue damage and periodontitis. Targeting neutrophil responses may help restore immune balance and prevent disease progression.

**Figure 3 microorganisms-13-02573-f003:**
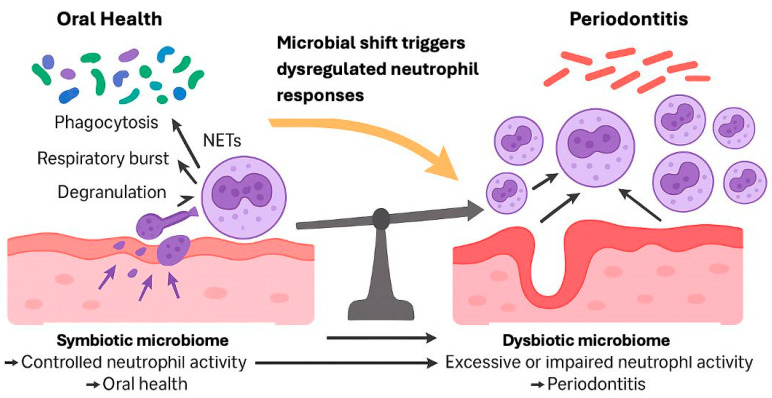
Microbial dysbiosis disrupts neutrophil homeostasis, leading to excessive or impaired responses. This dysregulation contributes to tissue damage and drives periodontitis progression.

## Data Availability

No new data were created or analyzed in this study. Data sharing is not applicable.

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
