# Peer review of "Neutrophils at the Crossroads of Oral Microbiome Dysbiosis and Periodontal Disease"

_microorganisms, 2025, doi:10.3390/microorganisms13112573_

Round 1

Reviewer 1 Report

Comments and Suggestions for Authors

As a major cellular compartment of the innate immune system, neutrophils play a major role in clearing microbial infections, but also in exacerbating inflammatory-diseases. Here, Viana and colleagues review the current literature around the role of neutrophils in these processes in the specific context of periodontal disease. The review is a comprehensive summary of the recent literature and is somewhat well organized. The major recent research is well-presented. The manuscript is divided into 6 sections: An introduction and conclusion, and 4 sub-headings. The review is a significant contribution to the field. What follows are some minor suggestions to improve the organization of the text.

MS is structured into Introduction, 4 subheadings, and Conclusion. Suggest to make all 4 subheading titles a question as in 1, 2 and 4. The title of sections 2 and 3 are too similar.

For example, change heading 2 to "Can neutrophils be used as a biomarker for periodontal disease severity?"

Suggest to swap the order of section 3 and 4. biomarker is more of a translational potential and fundamentally different from the basic biology presented in sections 1, 2, and 4. Alternatively, re-work section 4 more as therapeutic approaches. This is more in line with the introduction and conclusion structure.

LL169-171 Some notes from authors accidentally left in the ms.

Author Response

We would like to thank the reviewer for the constructive appraisal of the manuscript and for the insightful comments and suggestions provided. We carefully considered all points and made the corresponding revisions to improve the clarity, coherence, and overall structure of the review. Below we provide a detailed response to each comment.

Comment #1:

MS is structured into Introduction, 4 subheadings, and Conclusion. Suggest to make all 4 subheading titles a question as in 1, 2 and 4. The title of sections 2 and 3 are too similar. For example, change heading 2 to “Can neutrophils be used as a biomarker for periodontal disease severity?”

Response:

We thank the reviewer for this helpful suggestion. We have revised the titles of the subheadings to ensure consistency in their format, making them all questions for greater clarity and engagement. In particular, Section 2 has been renamed to: “Can neutrophils be used as biomarkers for periodontal disease severity?” to clearly distinguishe it from Section 3 and align with the reviewer’s recommendation.

The updated titles can be found in the revised manuscript.

Comment #2:

Suggest to swap the order of section 3 and 4. Biomarker is more of a translational potential and fundamentally different from the basic biology presented in sections 1, 2, and 4. Alternatively, re-work section 4 more as therapeutic approaches. This is more in line with the introduction and conclusion structure.

Response:

We appreciate this insightful suggestion. We revised the structure accordingly to improve the logical flow of the manuscript. The sections were re-ordered to better reflect the progression from basic biological mechanisms to translational applications. Specifically, the section discussing biomarkers now precedes the section on therapeutic approaches, which has been reworked to better highlight emerging therapeutic perspectives related to neutrophil modulation in periodontal disease.

These structural and content adjustments improve the coherence between the introduction, main body, and conclusion, as recommended.

All changes are reflected in the revised version of the manuscript.

Reviewer 2 Report

Comments and Suggestions for Authors

The authors here seek to cover the role of neutrophils in oral dysbiosis in this review. The methods state that it is a literature review but there doesn't seem to be any cohesive methodology so I'm not sure that section is needed in the abstract. Abstract also seems to have some formatting issues.

Introduction: 

Figure seems to be original and well made but not labelled as 'figure'.

The authors only need to define acronyms once, ROS is definted several times, as are NETs.

Overall good content. First section is labelled intro but then second section is just some content- so there are no methods used here then it seems? 

ln 171 seems to be a note between the authors that should not be in the text. 

Second and third figures are good but also need labelling. Section 3 seems kind of meander at times without a clear goal which I think is something the text suffers from. The sections are not demarcated enough to help guide the reader through. There are also several sections that seem to have overused citations like in the 330s where [5] is used a lot. 
Conclusion is brief. 

Overall I think this is a decent summary but it needs better headings to be a bit more clear.

Author Response

We thank the reviewer for their constructive feedback and thoughtful comments that have helped us improve the clarity, structure, and presentation of our manuscript. We carefully revised the text to address each of the issues raised, as detailed below.

Comment #1:

The methods state that it is a literature review but there doesn’t seem to be any cohesive methodology so I’m not sure that section is needed in the abstract. Abstract also seems to have some formatting issues.

Response: Thank you for your remark. In fact, this narrative review follows similar reviews already published in this journal, upon similar formats. For example: https://www.mdpi.com/2076-2607/12/11/2341.

Comment #2:

Figure seems to be original and well made but not labelled as ‘figure’.

Response:

Thank you for pointing this out. All figures have now been properly labelled and numbered according to journal formatting guidelines.

Comment #3:

The authors only need to define acronyms once, ROS is defined several times, as are NETs.

Response:

We thank the reviewer for noting this. Repeated acronym definitions have been removed, and all abbreviations are now defined only upon their first appearance in the text.

Comment #4:

Overall good content. First section is labelled intro but then second section is just some content- so there are no methods used here then it seems?

Response: As previously responded in comment #1, this narrative review follows similar reviews already published in this journal, upon similar formats. For example: https://www.mdpi.com/2076-2607/12/11/2341.

Comment #5:

ln 171 seems to be a note between the authors that should not be in the text.

Response:

We thank the reviewer for identifying this oversight. The internal note has been removed from the revised version.

Comment #6:

Second and third figures are good but also need labelling.

Response:

As suggested, Figures 2 and 3 have now been properly labelled and formatted in accordance with the journal’s requirements.

Comment #7:

Section 3 seems kind of meander at times without a clear goal which I think is something the text suffers from. The sections are not demarcated enough to help guide the reader through.

Response: We have made changes upon other reviewers remarks and we hope they might see fit in a more clearer goal.

Comment #8:

There are also several sections that seem to have overused citations like in the 330s where [5] is used a lot.

Response:

We thank the reviewer for this observation. We understand the concern regarding repeated citations; however, we decided to maintain them in order to ensure proper attribution and academic accuracy, particularly where multiple statements are directly supported by the same source. The reference in question is considered a reference review paper with 183 references cited, and for this reason a comprehensive and broad narrative review work.

Comment # 9:

Conclusion is brief.

Response:

We appreciate the reviewer’s suggestion. We opted to keep the conclusion concise, as our goal was to provide a clear and focused summary of the main findings without unnecessary repetition of content already discussed in detail throughout the manuscript. This approach ensures clarity and maintains the reader’s attention on the key take-home messages.

Reviewer 3 Report

Comments and Suggestions for Authors

This paper nicely describes, in more or less a text book style, the role that neutrophils play in periodontal health and disease. It is not a review, but a general nice description of the potential roles of neutrophils in periodontitis. The description is correct, although misses any link to the role that periodontitis (and neutrophils in particular) might have in diseases related to periodontitis. Adding such information, even speculative, might enlarge the value of this ‘review’. E.g., the role of periodontitis and neutrophils in, e.g., diseases as rheumatoid arthritis, diabetes mellitus and inflammatory bowel diseases. It is not so interesting to focus on diagnostic and treatment approaches as their a clinical tools to detect periodontitis and monitor the treatment effect.

Author Response

We sincerely thank the reviewer for their thoughtful and constructive feedback, as well as for recognising the clarity and accuracy of our description of neutrophil function in periodontal health and disease.

Comment:

This paper nicely describes, in more or less a text book style, the role that neutrophils play in periodontal health and disease. It is not a review, but a general nice description of the potential roles of neutrophils in periodontitis. The description is correct, although misses any link to the role that periodontitis (and neutrophils in particular) might have in diseases related to periodontitis. Adding such information, even speculative, might enlarge the value of this ‘review’. E.g., the role of periodontitis and neutrophils in, e.g., diseases as rheumatoid arthritis, diabetes mellitus and inflammatory bowel diseases. It is not so interesting to focus on diagnostic and treatment approaches as their a clinical tools to detect periodontitis and monitor the treatment effect.

Response:

We appreciate the reviewer’s kind remarks and insightful suggestions. We acknowledge the value of linking neutrophil-mediated mechanisms in periodontitis with systemic inflammatory diseases such as rheumatoid arthritis, diabetes mellitus, and inflammatory bowel disease. However, we decided to keep the focus of this manuscript strictly on the role of neutrophils within the periodontal context, as broadening the scope to include systemic associations would require an additional level of discussion beyond the intended objectives of this review.

We believe that maintaining this focused perspective ensures a clear and cohesive narrative consistent with the manuscript’s aim to synthesise current evidence on neutrophil activity in periodontal health and disease. Nonetheless, we have briefly acknowledged the systemic implications of periodontal inflammation in the conclusion to highlight the relevance of this topic for future research.

Round 2

Reviewer 3 Report

Comments and Suggestions for Authors

The review has improved. However, I feel that the change in subtitle 2 is not correct and does not reflect the content of this section. The old subtitle 2 was much better. Furthermore, I remain to my previous comment that the link to the role that periodontitis (and neutrophils in particular) might have in diseases related to periodontitis is missing. It is only added to one sentence in the conclusion. I feel that adding a paragraph on the possible link to systemic diseases might increase the value of this review. The information on this possible link is eagerly missing. Such a paragraph, even speculative, might enlarge the value of this ‘review’. E.g., the role of periodontitis and neutrophils in, e.g., diseases as rheumatoid arthritis, diabetes mellitus and inflammatory bowel diseases.

Author Response

Comment #1

I feel that the change in subtitle 2 is not correct and does not reflect the content of this section. The old subtitle 2 was much better.

Answer #1.

We appreciate this feedback. In fact this change was due to a suggestion by another reviewer. Nevertheless we agree and have replaced with the original title.

Comment #2

Furthermore, I remain to my previous comment that the link to the role that periodontitis (and neutrophils in particular) might have in diseases related to periodontitis is missing. It is only added to one sentence in the conclusion. I feel that adding a paragraph on the possible link to systemic diseases might increase the value of this review. The information on this possible link is eagerly missing. Such a paragraph, even speculative, might enlarge the value of this ‘review’. E.g., the role of periodontitis and neutrophils in, e.g., diseases as rheumatoid arthritis, diabetes mellitus and inflammatory bowel diseases.

Answer #2.

We thank the reviewer for this constructive comment. Following this valuable suggestion, we have expanded the manuscript by adding a dedicated paragraph describing the potential mechanistic links between neutrophil dysfunction in periodontitis and systemic inflammatory diseases. The new text discusses how genetic and epigenetic predisposition, dysbiotic microbial stimulation, and exaggerated NET formation may contribute to endothelial dysfunction and low-grade systemic inflammation 
